# Design of Bionic Foot Inspired by the Anti-Slip Cushioning Mechanism of Yak Feet

**DOI:** 10.3390/biomimetics9050260

**Published:** 2024-04-25

**Authors:** Weijun Tian, Kuiyue Zhou, Zhu Chen, Ziteng Shen, Zhirui Wang, Lei Jiang, Qian Cong

**Affiliations:** 1Key Laboratory of Bionic Engineering, Ministry of Education, Jilin University, Changchun 130022, China; tianweijun@jlu.edu.cn (W.T.); zhouky1818@163.com (K.Z.); chenzhu21@mails.jlu.edu.cn (Z.C.); shenzt21@mails.jlu.edu.cn (Z.S.); 2North-Vehicle Research, Fengtai District, Beijing 100072, China; zhrwang@foxmail.com (Z.W.); leijiang@noveri.com.cn (L.J.)

**Keywords:** yak foot, non-slip cushioning, bionic foot, simulation

## Abstract

In recent years, legged robots have been more and more widely used on non-structured terrain, and their foot structure has an important impact on the robot’s motion performance and stability. The structural characteristics of the yak foot sole with a high outer edge and low middle, which has excellent soil fixation ability and is an excellent bionic prototype, can improve the friction between the foot and the ground. At the same time, the foot hooves can effectively alleviate the larger impact load when contacting with the ground, which is an excellent anti-slip buffer mechanism. The bionic foot end design was carried out based on the morphology of the yak sole; the bionic foot design was carried out based on the biological anatomy observation of yak foot skeletal muscles. The virtual models of the bionic foot end and the bionic foot were established and simulated using Solidworks 2022 and Abaqus 2023, and the anti-slip performance on different ground surfaces and the influence of each parameter of the bionic foot on the cushioning effect were investigated. The results show that (1) the curved shape of the yak sole has a good anti-slip performance on both soil ground and rocky ground, and the anti-slip performance is better on soil ground; (2) the curved shape of the yak sole has a larger maximum static friction than the traditional foot, and the anti-slip performance is stronger under the same pressure conditions; (3) the finger pillow–hoof ball structure of the bionic foot has the greatest influence on the buffering effect, and the buffering effect of the bionic foot is best when the tip of the bionic foot touches the ground first.

## 1. Introduction

Legged robots have received much attention in the past few years due to their excellent performance in complex environments [1,2,3]. And the foot structure is one of the important factors affecting the walking performance of quadruped robots [4,5,6,7,8]. The foot mainly plays the role of cushioning and support, and exhibits unique functions and properties during robot activities, which need to be adapted to different types of ground. Most importantly, the anti-slip performance depends largely on the structure of the foot, which plays a decisive role in the stable walking of the robot [9,10,11,12,13]. The development of bionics provides new research ideas for the foot design of robots and improving the locomotion of quadruped robots [14,15].

The foot of an animal consists of many bones, muscles, ligaments, and joints, and is the main component of weight bearing and locomotion. Due to the special organizational structure and biomechanical properties [16], the foot mainly plays the role of padding and support and exhibits unique functions and properties in animal activities. During movement, the foot is the only part of the animal that is in contact with the ground, and the instantaneous force is several times greater than the weight [17,18] due to the sudden change in force at the time of contact. The adaptation of these animals to rugged ground is largely due to the protection of their foot structures. Inspired by the fact that animals can adapt to rugged ground and their feet show excellent anti-slip, researchers have designed a series of bionic foot ends and proposed a variety of methods to absorb the impact energy [19]. Compared with regular foot ends, such as spherical foot ends and cylindrical foot ends, bionic foot ends have the advantages of stronger anti-slip and better adaptability to the terrain.

In different environments, the functionalities required for robot legs vary. Researchers seek inspiration from various organisms, each providing unique solutions to specific challenges. Domestic and foreign studies indicate that different biomimetic foot designs can enhance a robot’s adaptability to non-structured terrain, ground grip capability, and cushioning performance [20,21,22,23,24,25,26,27]. By mimicking the irregular shape of a cat’s paw, the “cheetah cub” quadruped robot can move swiftly on rugged terrain [20]; through the study of the camel’s unique biological structure, camel-inspired mechanical legs exhibit strong adaptability to sand and soft soil [21]; and utilizing 3D scanning and optimization techniques on ostrich hind limbs, the design of biomimetic mechanical legs demonstrates strong adaptability to sandy environments [22]. Current bio-inspired foot designs can adapt well to various terrains, but their performance in complex environments may be limited due to a lack of dynamic adjustment capability, prompting researchers to further study self-adaptive mechanisms. Research on the ground gripping ability of bio-inspired feet is also an important aspect continually being improved by researchers. The bio-inspired gecko robot “Stickybot” offers a design of underactuated multi-material structures that conform to surfaces ranging from centimeters to micrometers in length, effortlessly adhering and detaching using millions of tiny hairs on the surface [23]; through histological observations of locust foot structures, a petal-shaped bio-inspired foot is designed, greatly enhancing the robot’s ground gripping ability [24]; and inspired by climbing mammals and describing the morphology of mammalian feet, a high-adhesive bio-inspired climbing shoe is designed [25]. Through continuous improvements by researchers, although it is difficult to provide consistent ground grip on different terrains, the gripping ability of robot legs has been greatly enhanced. There has been significant progress and improvement in both the adaptability of robot legs to non-structured terrain and research on the ground gripping ability. As the part of the body directly in contact with the ground, feet generate significant impact forces during movement and withstand considerable external stimuli, greatly affecting the overall speed and stability of the robot. Research on the cushioning function of bio-inspired feet can effectively mitigate external stimulus loads and play a crucial role in shock absorption and support. Therefore, researchers have conducted a series of studies on the cushioning performance of bio-inspired feet. Bio-inspired robot feet based on the skeletal structure of German Shepherds can convert rigid contact between the robot and the ground into flexible contact, reducing vibrations generated by the ground impact [26]; bio-mechanical feet designed based on the movement characteristics and morphology of goats play an anti-slip and cushioning role through joint adjustments [27]. Currently, there are still some limitations in the research on the cushioning performance of bio-inspired feet, as the cushioning structures in bio-inspired feet need to endure long periods of repeated compression, deformation, and wear without losing effectiveness. Therefore, researchers are introducing new bio-inspired designs to enhance the shock-absorbing ability of bio-inspired feet while maintaining their durability and performance. Different studies on bio-inspired feet with different functionalities not only propose innovative solutions but also emphasize the importance of bio-inspired design in addressing complex engineering challenges and advancing robotics development.

It is widely recognized that over 50% of the Earth’s land surface comprises rugged hills or low-lying wet swamps [28]. When the need arises for robots to traverse tight spaces [29] or navigate through unstructured terrain (rugged mountains, etc.) [30], to execute tasks including handling [31], disaster prevention [32], rescue [33], and exploration [34] in non-structural terrains, the challenges are considerable. Legged robots, on the other hand, show the advantages of flexibility and terrain adaptability due to their structural characteristics [35]. As a typical non-structural terrain, the high mountain plateau is extremely difficult for humans to work in this environment due to its harsh geographic, geological, and climatic conditions. Therefore, it is of practical significance to develop robots that can perform unmanned surveying, material transportation [36], and other tasks to replace humans in complex and harsh environments. Moreover, the harsh environment poses formidable challenges to the performance of heavy-duty robots, with the foot structure of the robot exerting a significant influence on their ability to bear heavy loads and move efficiently. Yak is a quadrupedal mammal that has lived in the plateau environment for a long time and has strong adaptability to the complex environment in the plateau region [37], with a large body size but excellent locomotor ability, which is closely related to its unique foot structure. The yak foot has two main hooves and two suspensory hooves, and the hooves play an important role in its locomotion [38]. And the hoof capsule, as an important part in direct contact with the ground, can instantly withstand a huge external impact, absorb the impact, and effectively reduce the vibration caused by the impact [39,40,41]. In this study, the yak hoof was taken as the research object, and the yak foot was dissected by an anatomical method and its histology was observed. Then, the biomechanical function of the bionic foot was simulated by using the finite element method (FEM), and the anti-slip performance of different feet was analyzed and compared; furthermore, a bionic mechanical foot was designed and the effect of each parameter on the cushioning performance was analyzed. This study provides a new idea and method for the design of an anti-slip cushioning foot end for quadruped robots.

## 2. Materials and Methods

### 2.1. Ethical Declaration

This study was approved by the Ethics Committee of Jilin University (Changchun, China).

The test yak feet and hooves were purchased from yak farms in Hongyuan County, Aba Prefecture, Sichuan Province, and two front and back hooves of adult yaks of 3 to 4 years old that were properly vaccinated, healthy, and normally slaughtered were selected for the observation of yak plantar morphology. One anterior and posterior foot of each yak was dissected below the carpal (forelimb) and tarsal (hindlimb) joints.

### 2.2. Anatomical Observations of Yak Foot

The yak foot was dissected in order to provide inspiration for the bionic model design. By separating the skin, as shown in Figure 1a, the tendons of the yak foot were observed (Figure 1b).

In order to observe the bone and tissue structure of the yak foot more comprehensively, another yak foot was taken (Figure 1c), and one of the two main fingers of the yak foot was selected and sawed along the longitudinal direction right in the middle of the hoof bone, coronoid bone, and phalanx bone of this finger (Figure 1d).

From Figure 1d, the bones of the yak foot were, from top to bottom, the metacarpal, phalanges, coronary bone, and hoof bone. The metacarpal and phalanges form the phalangeal joint, and the shape of the bones limits the phalangeal joint to extension and flexion movements, and the lateral collateral ligaments on both sides limit its range of motion. Near the phalangeal joint are four seed bones, each of which forms an articulation with the articular surface of each metacarpal bone, anchored in position by the lateral collateral ligament of the seed bone and the distal collateral ligament of the seed bone and assisting in the support of this joint. The phalanges form a coracoid joint with the coracoid bone, which can also only move in extension and flexion, with the lateral collateral ligament and the palmar collateral ligament in the vicinity of the joint. The hoof bone forms a hoof joint with the coronoid bone, which is encased in a hoof box and has an extremely limited range of motion. The articular surface of the hoof joint also has an inferior seed bone on the palmar side of the joint, which is held in place by the suspensory ligament extending from the coronoid bone.

The movement of the yak foot is mainly realized by the extension and contraction of many tendons in the foot, among which the extensor tendons on the dorsal side of the foot (Figure 1e) and the flexor tendons on the medial side of the foot (Figure 1f) are the main controllers. Three extensor tendons were clearly seen on the dorsal side of the yak foot, which together controlled the extension movement of the yak foot. The three extensor tendons extended from the upper end of the metacarpal bone all the way down to the upper end of the hoof bone. The medial tendons of the foot included three flexor tendons, which controlled the flexion of the yak foot, starting from above the metacarpal bone and ending at the palmar surface of the hoof bone.

### 2.3. Bionic Foot End Model Construction and Simulation

Medical image data acquisition of the yak foot was performed using computed tomography (CT) and magnetic resonance imaging (MRI), in which the experimental equipment was the Activion series 16-slice multi-row spiral CT scanner from Toshiba, Japan, and the Lianyin 3.0 T magnetic resonance imaging system from Shanghai Lianyin Medical Technology Co., Shanghai, China. The skeletal structure and shape model of the yak foot were extracted using Mimics, and the shape model of the yak hoof was created in Geomagic for reconstruction and fitting of the plantar surface (Figure 2). 

Furthermore, a finite element model based on the organizational structure of the yak hoof was established and used to investigate the biomechanical function of this structure. A finite element model of the yak hoof was built by the model reconstruction of the plantar morphology of the yak (Figure 3a), and the common hemispherical foot ends of appropriate sizes (Figure 3b), cylindrical foot ends (Figure 3c), and the ground model (Figure 3d) were also built. The reason why we established the finite element model of the cylindrical foot and hemispherical foot was that these two types of foot are the most widely used in quadruped robots, and they have their own advantages, with the cylindrical foot having a strong load-bearing capacity and the hemispherical foot having a strong adaptability to the terrain but being prone to slipping. On the basis of this, we scanned the plantar morphology of biological prototypes and observed the biophysical structure to obtain design inspiration and adopted the biomimetic design method, aiming at designing a biomimetic foot with both anti-slip performance and load-carrying capacity.

A 3D model involving all the parts of the bionic model was built and ground on the FEM software ABAQUS 2023. The process of each foot end contacting the ground was simulated through the dynamic simulation and analysis of ABAQUS 2023, and at the same time, different ground conditions were simulated by changing different performance parameters of the ground model for comparative analysis. At present, the foot end of quadruped robots is often made of elastic materials such as rubber and the interaction between the foot end and the ground is similar to the extrusion and friction between car tires and the ground. Therefore, in the finite element analysis, the hard rubber commonly used on the surface of tires was chosen as the material of the foot end, and the material properties are shown in Table 1, the performance parameters of different ground (soil, rock) are shown in Table 2, and the friction coefficients between the foot end and different ground are shown in Table 3. The mesh size of the finite element method was set to 3 mm.

In the process of contact between the foot end and the ground, the maximum static friction between the foot end and the ground was used as a measure of the anti-slip performance. In the case of giving the same positive pressure on the foot end, applying a horizontal displacement load to it in the horizontal direction to make the foot end slide, the larger the maximum static friction was, the better the anti-slip performance of the foot end was. The contact setting is shown in Figure 4, and the lower surface of the hemispherical foot end and cylindrical foot end were relatively regular, so the lower surface of the foot end was selected to contact, and the lower surface of the bionic foot end was not regular, so the universal contact was selected to let the foot end find the contact position with the ground automatically; the load setting is shown in Figure 5, and the concentrated load in the vertical direction was set in the reference point, which was taken as 50 N, 100 N, 150 N, and 200 N for the analysis, and the X-ray displacement load was applied in the horizontal direction, so that the maximum static friction was higher. The condition of 200 N was analyzed, while a linearly increasing horizontal displacement load was applied to the X-positive direction to move the foot end part to calculate the frictional resistance.

### 2.4. Bionic Foot Model Construction and Simulation

The design of the bionic foot was based on the structure of the yak foot and the main structures that played a role in the cushioning process. The bones were simplified as rods, the elastic structures such as tendons, phalanges, and hoof balls were designed as springs, the dimensions of the structures were referenced to the proportion of the actual size of the yak foot, and the overall symmetrical structure was adopted. Solidworks 2022 was used for the design and modeling of the bionic foot (Figure 6).

The bionic foot was mainly composed of a metacarpal structure, a phalangeal structure, a coracoid–seed bone structure, a hoof bone structure, flexor and extensor tendon springs, and a finger occipital–hoof ball spring. The upper end of the metatarsal structure connected to the leg structure, while the lower end connected to the phalangeal structures of the third and fourth fingers via two rotating subsets, forming joint 1. Joint 2 formed between the phalangeal structure and the coracoid–seed bone structure. The coracoid–seed bone structure, shaped like an L, mimics the crown bone–seed bone structure, simulating the compression of the lower seed bone after the crown bone’s force. Joint 3 formed between the corners of the L-shaped structure and the hoof bone structure. A horizontal groove facilitated connection with the finger pillow–hoof ball structure, which mainly comprised a compression rod, spring, and guide bar. The combination of the crown bone–seed bone structure and the finger pillow–hoof ball structure mimicked the force exerted on the crown bone by the tethered bone, compressing the seed bone and the elastic components such as the finger cushion and hoof ball, resulting in their compression and deformation. This mechanism served as the primary buffering structure of the bionic foot. Figure 7 illustrates the cross-section schematic of the finger cushion–hoof ball cushioning structure.

The simplified model was imported into ABAQUS 2023 after removing non-essential parts and springs. The bionic foot was made of aluminum alloy, and the material properties are shown in Table 4, the corresponding springs were added to ABAQUS 2023 and the dynamics of the bionic foot were simulated, and the mesh size of the finite element method was set to 3 mm. In order to obtain the cushioning performance of the bionic foot with different spring parameters and explore the effect of the spring parameters on the cushioning performance, the time taken when the velocity of the metacarpal structure decreases to 0, the impact time was compared. Additionally, the impact velocity of the metacarpal structure was set to 500 mm/s to analyze and compare the time required for its velocity to reach 0 under different spring parameters. The longer the impact time, the better the cushioning effect of the mechanism. Simulate the impact of the bionic foot under the three states of 5°, 0°, and −5° angle between the bottom surface of the bionic foot and the ground, analyze the influence of the parameters of the bionic foot on the cushioning effect of the degree of which the test factors were selected as extensor tendon spring stiffness coefficient z1, the spring parameter of the finger occiput–hoof ball structure z2, and the spring stiffness coefficient of the flexor tendon z3, and the level of the test factors is shown in Table 5.

## 3. Result and Analysis

### 3.1. Anti-Slip Properties of the Foot End

Since the deformation of the soil ground was greater during the action of the foot end with the ground, it was more favorable to observe the action of the three kinds of foot ends, namely, the bionic foot end, hemispherical foot end, and cylindrical foot end, with the ground components; therefore, under the same pressure conditions, the topography of the soil was selected to observe the stress clouds of the three kinds of foot ends (Figure 8). It can be seen from the figure, for the three kinds of foot end and soil action, the soil surface has different degrees of deformation, the hemispherical foot end stress map is circular, and the stress is mainly concentrated in the center of the circle; the cylindrical foot end stress map is capsule-shaped, the stress is mainly concentrated in the capsule at both ends of the capsule shape; the bionic foot end stress map is hoof-shaped, is mainly distributed in the hoof weight-bearing surface and hoof ball parts, and the stress is concentrated in the hoof ball parts.

Figure 9 shows the trend of the friction between the three kinds of foot ends and the soil ground under different pressures. From Figure 9, it can be seen that the friction between the three kinds of foot ends and the ground increases with the increase in the displacement load and starts to slide after reaching the maximum static friction. From Figure 9a, under the condition of 50 N pressure, the hemispherical foot end, cylindrical foot end, and bionic foot end slip at 0.6 s, 0.4 s, and 0.7 s, and the maximum static friction is 12.5 N, 12.2 N, and 14.1 N. From Figure 9b, under the condition of 100 N pressure, the hemispherical foot end, cylindrical foot end, and bionic foot end slip at 0.6 s, 0.4 s, and 0.75 s, and the maximum static friction is 25.2 N, 24.1 N, and 30.5 N. From Figure 9c, under the pressure condition of 150 N, the hemispherical foot end, cylindrical foot end, and bionic foot end slide at 0.6 s, 0.4 s, and 0.8 s, and the maximum static friction is 38.0 N, 37.0 N, and 47.1 N. From Figure 9d, it can be seen that under the condition of 200 N pressure, the hemispherical foot end, cylindrical foot end, and bionic foot end slip at 0.6 s, 0.4 s, and 0.7 s, with a maximum static friction of 49.4 N, 49.3 N, and 62.9 N.

The maximum static friction between the foot end and the ground increased with the increase in the vertical pressure on the foot end. Further analysis revealed that the relationship between the maximum static friction and pressure of the hemispherical foot end and the cylindrical foot end and the soil ground conforms to the friction formula fs = μN, while the friction of the bionic foot end and the soil ground did not conform to it. The maximum static friction was higher than the theoretical value in the conditions of 50 N, 100 N, 150 N, and 200 N, which were 12.8%, 22%, 25.6%, and 25.8% higher than the theoretical value, respectively. The analysis, combined with the stress cloud diagram in Figure 8, indicates that due to the soft soil, the foot end compresses the ground under vertical pressure, causing it to sink. The special concave structure at the bottom of the bionic foot end acts as a soil fixation mechanism, restricting the soil flow and generating extra friction resistance, thus preventing slipping. By increasing the vertical pressure, more soil is restrained by the bottom structure, increasing extra friction. And as the vertical pressure increased, the more the soil was restricted by the bottom structure of the foot, the more frictional resistance was generated, and when the pressure continued to increase, the void at the bottom of the foot was filled, and the additional frictional resistance reached its maximum value. Therefore, the bionic foot end exhibited a superior anti-slip performance compared to the hemispherica and the cylindrical foot end on soil surfaces.

Similarly, Figure 10 shows the trend of the friction change between the three foot ends and the rocky ground under different pressures. In the rocky ground, the trend of the friction changes was consistent with the soil ground; from Figure 10a, under the condition of 50 N pressure, the hemispherical foot end, cylindrical foot end, and bionic foot end slipped at 0.6 s, 0.5 s, and 0.65 s, and the maximal static friction was 19.8 N, 20.1 N, and 21.5 N. From Figure 10b, it can be seen that under the condition of 100 N pressure, the hemispherical foot end, cylindrical foot end, and bionic foot end slipped at 0.55 s, 0.45 s, and 0.6 s, and the maximal static friction was 41 N, 40 N, and 45 N. From Figure 10c, under the pressure condition of 150 N, the hemispherical foot end, cylindrical foot end, and bionic foot end slipped at 0.65 s, 0.6 s, and 0.7 s, and the maximal static friction was 61 N, 60 N, and 65 N. From Figure 10d, it can be seen that under the pressure condition of 200 N, the hemispherical foot end, cylindrical foot end, and bionic foot end slipped at 0.65 s, 0.5 s, and 0.7 s, and the maximum static friction forces were 79.6 N, 79.4, and 85.1 N. 

Despite the minimal deformation of the rocky ground and the limited solidifying effect of the bionic foot end, its unique plantar morphology enabled it to maintain good contact with the ground during action on rocky terrain, thus preventing slipping. Consequently, the maximal static friction of the bionic foot end was still bigger than that of the hemispherical and cylindrical foot end under the same pressure condition. While the anti-slip performance of the bionic foot end on rocky ground may not match that on soil ground, it still demonstrated effective anti-slip capabilities.

### 3.2. Bionic Foot Cushioning Properties

ABAQUS 2023 was used to simulate and analyze the bionic foot, and output the velocity change in the metacarpal bone structure in each group of tests under the three touchdown conditions (Figure 11). The results of the tests were tabulated, as shown in Table 6, and the slope of the curve is the acceleration of the metacarpal bone during its movement. The observation of the curves reveals that, in the moment of touchdown, the acceleration of the metacarpal bone structure of the bionic foot was not large, but it was slowly increasing over time, which indicates that the bionic foot could relieve the huge impact force when touching the ground. From Figure 11a, when the angle between the sole of the foot and the ground is 0°, the impact time of test No. 4 was the shortest, 0.0201 s, and that of test No. 1 was the longest, 0.0329 s; from Figure 11b, when the angle between the sole of the foot and the ground is 5°, the impact time of test No. 9 was the shortest, 0.0323 s, and that of test No. 1 was the longest, 0.0474 s; from Figure 11c, it can be seen that when the angle between the sole of the foot and the ground is −5° for impact, the impact time of test No. 3 was the shortest, 0.0199 s, and the impact time of test No. 1 was the longest, 0.0348 s.

The bionic robot foot exhibited optimal cushioning when tested under condition 1, with the extensor tendon spring stiffness, finger occiput–hoof ball spring, and flexor tendon spring all set at 50 N/mm. The lower the spring stiffness, the better the cushioning performance, within the limits of the structure. Additionally, the best cushioning effect occurred when the sole angle to the ground was 5° during impacts, aligning with the scenario where the toes make initial contact with the ground during the yak movement.

### 3.3. Regression Analysis

A partially orthogonal polynomial regression design was utilized to seek equations for the relationship between the spring stiffness coefficients and impact time for each of the three tests. When the angle between the foot sole and the ground is 0°:(1)$$\left\{ \begin{array}{l} S=\sum_{1}^{9}y_{i}^{2}-\frac{1}{9}{\left(\sum_{1}^{9}y_{i}\right)}^{2}=139.856\\ f=9-1=8\\ f_{H}=3\\ S_{H}=135.959\\ S_{R}=S-S_{H}=3.897\\ f_{R}=5 \end{array}\right.$$

The test had no replicated trials to estimate the sum of squared errors, and the values $S_{R}$ were much smaller than $S_{H}$ and $S_{R}/f_{R}$ would therefore be used as the test error estimates.
(2)$$F_{H}=\frac{S_{H}/f_{H}}{S_{R}/f_{R}}=\frac{135.959/3}{3.897/5}=58.147>F_{0.01}(3,5)=12.06$$

Loss-of-fit test using the contribution of the residual sum of squares:(3)$$\beta_{R}=\frac{S_{R}}{S}\times 100\%=2.786\%$$

More than 97% of the fluctuations in the test indicators were caused by $S_{H}$. Therefore, the regression equation was considered to be non-discrepant. The regression equation for the coding space could be derived as follows:(4)$$\hat{y}=0.024378-1.117\times 10^{-3}X_{1}(z_{1})-4.233\times 10^{-3}X_{1}(z_{2})-1.783\times 10^{-3}X_{1}(z_{3})$$

The regression equation in the coded space was changed to the regression equation in the natural space by the variation equation:(5)$$\left\{ \begin{array}{l} X_{1}(z_{1})=\psi_{1}(z_{1})=\frac{z_{1}-\overset{-}{z_{1}}}{\Delta_{1}}=\frac{1}{50}z_{1}-2\\ X_{1}(z_{2})=\psi_{1}(z_{2})=\frac{z_{2}-\overset{-}{z_{2}}}{\Delta_{2}}=\frac{1}{50}z_{2}-2\\ X_{1}(z_{3})=\psi_{1}(z_{3})=\frac{z_{3}-\overset{-}{z_{3}}}{\Delta_{3}}=\frac{1}{50}z_{3}-2 \end{array}\right.$$

The regression equation for natural space is obtained by bringing Equation (5) into Equation (4):(6)$$\hat{y}=0.04186-2.234\times 10^{-5}z_{1}-1.619\times 10^{-4}z_{2}-3.566\times 10^{-5}z_{3}$$

The regression equation for natural space could be obtained in the same way when the sole of the foot is at 5° to the ground:(7)$$\hat{y}=0.05492-7.1\times 10^{-5}z_{1}-6.934\times 10^{-5}z_{2}-3.2\times 10^{-5}z_{3}$$

When the sole of the foot was at −5° to the ground, the regression equation for natural space:(8)$$\hat{y}=0.04081-1.834\times 10^{-5}z_{1}-8.8\times 10^{-5}z_{2}-5.0\times 10^{-5}z_{3}$$

From Equations (6) and (8), it can be seen that the impact time and z1, z2, and z3 were negatively correlated; the impact time with the reduction in the three spring parameters increases, and the finger occipital–hoof ball spring stiffness had the greatest effect on the impact time, followed by the flexor tendon spring stiffness, and the extensor tendon spring stiffness had the least effect, indicating that the foot sole and the ground at 0° and −5° for the impact of the finger occiput–hoof ball structure played a major buffering effect. From Equation (7), we can see that the rule of change is similar to that of 0° and −5°, the stiffness of the extensor tendon spring and the stiffness of the phalanx–soleus spring had a large effect on the impact time, and the degree of effect was not much different, indicating that the extensor tendon spring and the phalanx–soleus spring both played a cushioning effect when the sole of the foot and the ground were impacted at a 5° angle.

When analyzing the three cases together, the impact time was negatively correlated with the stiffness of all three types of springs. Under the three touching angles, the bionic robot foot’s finger pillow–hoof ball structure had a greater influence on the buffering effect, and the bionic robot foot had the best buffering effect when the bionic foot touched the ground first, which was in line with the mechanism that the actual yak foot touched the ground with the toes first, and then buffered the yak foot by pressing the finger pillow and hoof ball.

## 4. Discussion

Alpine and highland areas boast abundant resources, yet they are situated in terrain that poses challenges for human exploitation. In comparison to human labor, legged robots exhibit advantages in flexible movement and robust terrain adaptability when undertaking tasks such as handling [30], disaster prevention [31], rescue [32], and exploration [33] on unstructured terrain, while the structure and performance of the foot end, which is the part of footed robots that is in direct contact with the ground, determines the stability of the footed robot’s movement and loading capacity. With the exploration and development of the plateau area, the operating environment puts forward new requirements for foot-type robots, especially the requirements of high load bearing, non-slip stability, and impact resistance. As a quadrupedal mammal living in the plateau environment for a long time, the yak has strong adaptability to the complex environment in the plateau area, can move quickly and stably in such a complex environment, and has excellent athletic ability in spite of its large size, which is closely related to its unique foot structure. Therefore, in this paper, the yak foot is selected as a bionic prototype, and a new type of bionic foot is designed to innovate the foot structure of quadrupedal robots, which improves the walking stability and buffering ability of quadrupedal robots.

The designed bionic foot mimicked the physical structure of a yak’s foot, and the bionic foot was constructed as a whole by connecting rods and springs. Unlike the bionic foot described in this paper, the foot ends of most quadrupedal robots are predominantly spherical and cylindrical. There were also some quadrupedal robots with irregularly shaped foot ends to realize some foot functions, such as “Cheetah-cub”, which could move forward quickly and smoothly on uneven surfaces [17], “Stickybot”, which realized the adsorption to the terrain by improving the material [18], the bionic foot with a petal structure designed by imitating the morphology of biological soles that could improve the gripping performance [19], and the bionic foot that could realize a certain degree of cushioning by adjusting the joints in combination with the movement characteristics and morphology of the goat [27]. The bionic foot presented in this paper was based on scans of yak plantar morphology and integrates features from the yak foot’s physical structure. Leveraging the cushioning function enabled by joint adjustments, a novel finger pillow–hoof ball structure was proposed to attenuate impacts through the compression and deformation of elastic components, thereby enhancing the cushioning performance. Incorporating such elastic structures into future articulated foot ends holds potential for improving foot functionality.

In this paper, we took the yak foot as the research object, established the yak plantar biological model, reflected the stress concentration through the stress cloud diagram, and compared it with the commonly used foot end of quadrupedal robots under different terrain conditions, which successfully proved the conclusion that the bionic foot end possesses stronger anti-slip performance, and also verifies that the plantar morphology feature of the yak foot really possesses excellent anti-slip characteristics, and it was a kind of excellent bionic blueprint. Previous studies have scanned and reconstructed the plantar morphology of reindeer [42] and ostrich [43] in three dimensions, and numerical simulation methods have found that their morphological structures have good anti-slip properties during foot–ground contact, which is consistent with the results of this paper, and both of them have proved that the ends of the feet of the biomimic have excellent anti-slip properties. We discussed the influence of each parameter of the bionic foot on the cushioning effect through the orthogonal experimental design method and proved that the bionic foot phalanx–hoof ball structure had the greatest influence on the cushioning effect. The impact of the hoof ball structure on the cushioning performance was also substantiated, with previous studies elucidating the cushioning mechanism of goat hoof balls [40]. Additionally, some studies had demonstrated that footpads could store and absorb mechanical forces [44], playing a crucial role in cushioning. These findings provided theoretical foundations for innovating foot designs of robots and improving the foot cushioning performance. Furthermore, it was verified that when the toe touched the ground first, the cushioning effect of bio-inspired feet was optimal, consistent with the movement of yaks in the natural world and aligning with the actual cushioning mechanism of yak feet. Previous studies on the cushioning mechanisms of bio-inspired robots had also confirmed this point [14].

However, there are still some limitations in this paper, such as this paper used reverse engineering to obtain the yak foot model and finite element analysis to study the anti-slip performance of the yak plantar surface, but it was not produced for the physical test; the design of the bionic foot could be further improved, and the performance of the bionic foot can be further improved from the perspective of materials based on the structural design; and only the anti-slip performance was compared with that of the most common hemispherical end and the cylindrical end, not with the rest of the bionic foot. Based on the study of the hooves of large animals, future work can explore the functions of the feet of small animals, compare and analyze them with existing studies, further optimize the bionic design by adding more bionic elements, and apply them to different conditions of movement, such as high load-bearing movement, high dynamic movement, etc., in order to enhance the universality of the quadrupedal robotic foot end. Therefore, extending the quadruped robot foot end through bionic structural design, materials, and other directions has great potential to further improve the physical limits of quadruped robots.

## 5. Conclusions

By dissecting the yak foot and observing its histology, the biological characteristics of the yak foot were studied. And a bionic mechanical foot was designed using the yak foot as a bionic prototype. The simulation results showed that the curved shape of the yak-like foot sole had a good anti-slip performance on both soil and rocky ground, the anti-slip performance was better on soil ground, and the curved shape of the yak-like foot sole had a stronger anti-slip performance compared with that of the traditional foot end. The simulation analysis of the cushioning characteristics of the bionic foot showed that the finger pillow–hoof ball structure of the bionic foot had the greatest influence on the cushioning effect, and the cushioning effect of the bionic foot was the best when the toe of the bionic foot touched the ground first, which was in line with the cushioning mechanism of the actual yak foot. The study and simulation analysis of the biological characteristics of the yak foot provide a new idea for the design of a bionic foot with an anti-slip cushioning function.

## Figures and Tables

**Figure 1 biomimetics-09-00260-f001:**
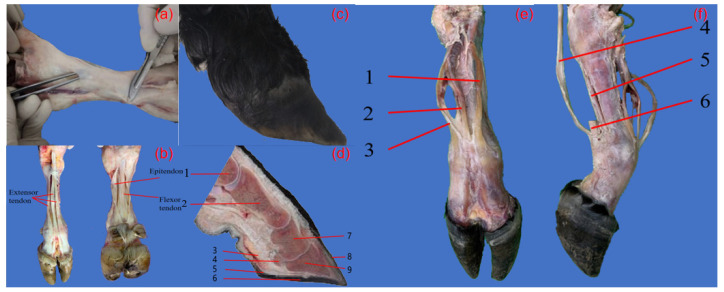
Yak foot anatomy. (**a**) Skin and fascia removal; (**b**) yak foot tendon; (**c**) another yak foot; (**d**) longitudinal section of yak foot, 1. Metacarpal, 2. Phalanges, 3. Phalangeal occiput, 4. Hypoglossal bone, 5. Hoof ball, 6. Hoof sole, 7. Coronary bone, 8. Hoof wall, 9. Hoof bone; (**e**) extensor tendon group; (**f**) flexor tendon group.

**Figure 2 biomimetics-09-00260-f002:**
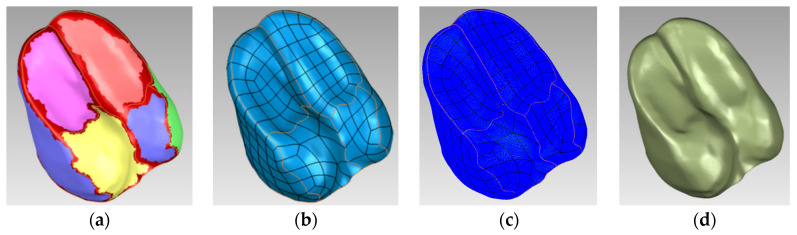
Three-dimensional modeling in Geomagic Studio. (**a**) Contour line detection stage; (**b**) surface sheet construction stage; (**c**) raster construction stage; (**d**) surface fitting stage.

**Figure 3 biomimetics-09-00260-f003:**
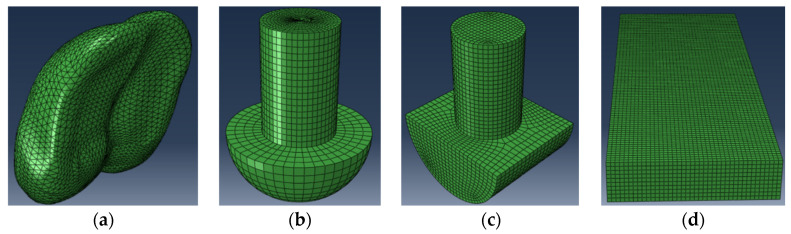
Each finite element model. (**a**) Finite element model of the yak hoof; (**b**) finite element model of the hemispherical foot end; (**c**) finite element model of the cylindrical foot end; (**d**) finite element model of the ground.

**Figure 4 biomimetics-09-00260-f004:**
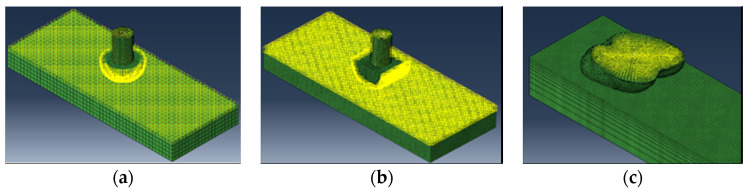
Contact setting for each foot end. (**a**) Contact setting for hemispherical foot end; (**b**) contact setting for cylindrical foot end; (**c**) contact setting for bionic foot end.

**Figure 5 biomimetics-09-00260-f005:**
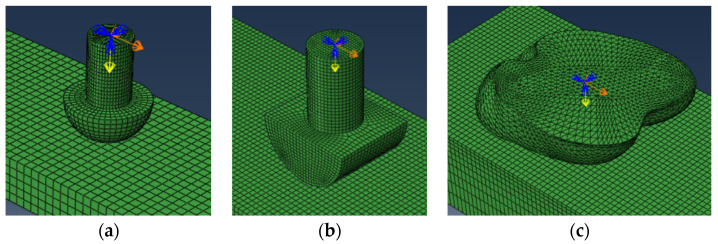
Load setting for each foot end. (**a**) Load setting for hemispherical foot end; (**b**) load setting for cylindrical foot end; (**c**) load setting for bionic foot end.

**Figure 6 biomimetics-09-00260-f006:**
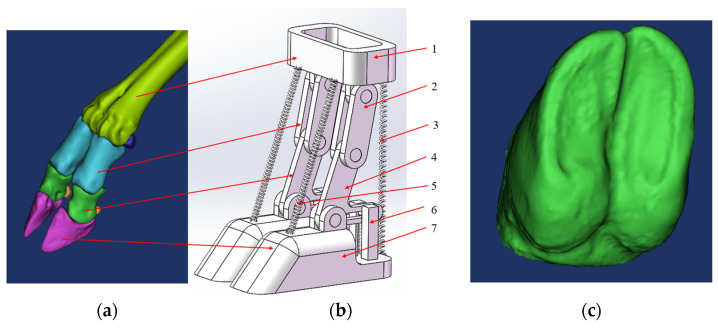
(**a**) Yak foot skeleton; (**b**) bionic foot. 1. Metacarpal structure; 2. Phalangeal structure; 3. Flexor tendon spring; 4. Coracoid–seed bone structure; 5. Extensor tendon spring; 6. Finger occipital–hoof ball spring; 7. Hoof bone structure; (**c**) the lower part of the foot bone structure.

**Figure 7 biomimetics-09-00260-f007:**
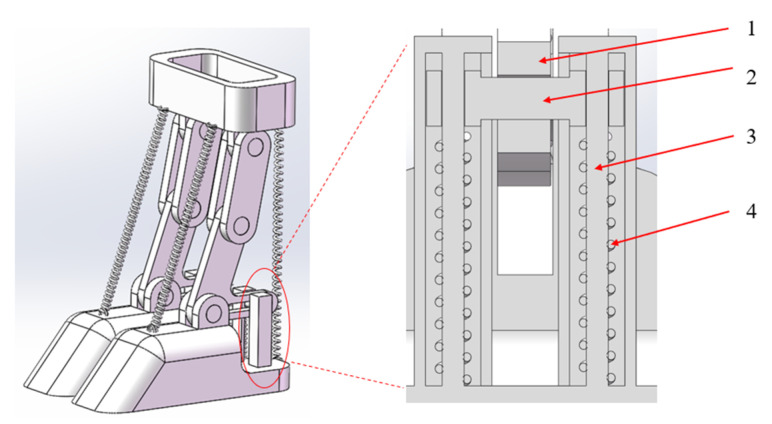
Finger cushion–hoof ball cushioning structure cross-section schematic. 1. Coracoid–seed bone structure; 2. Compression rod; 3. Guide bar; 4. Spring.

**Figure 8 biomimetics-09-00260-f008:**
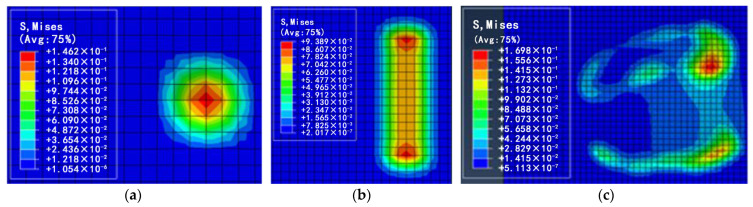
Stress cloud diagrams at the soil surface for each foot end. (**a**) Stress cloud diagrams for hemispherical foot end; (**b**) stress cloud diagrams for cylindrical foot end; (**c**) stress cloud diagrams for bionic foot end.

**Figure 9 biomimetics-09-00260-f009:**
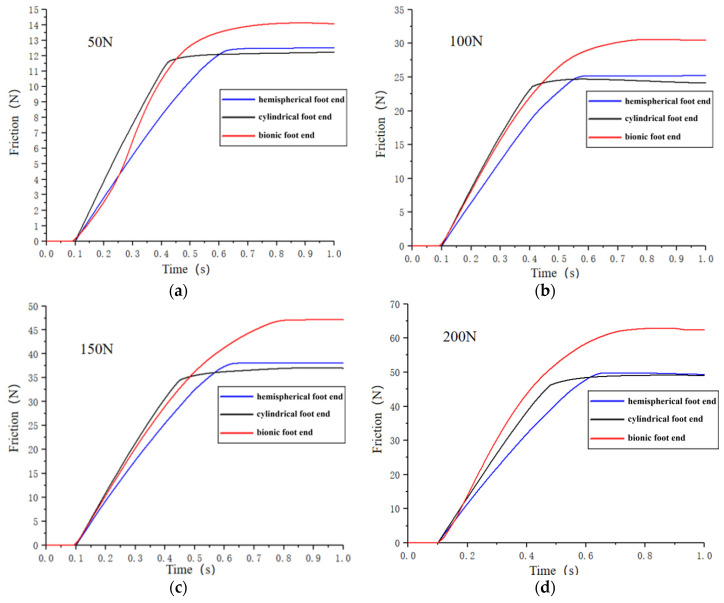
Friction between three types of foot ends and the soil ground at different pressure. (**a**) Pressure of 50 N; (**b**) pressure of 100 N; (**c**) pressure of 150 N; (**d**) pressure of 200 N.

**Figure 10 biomimetics-09-00260-f010:**
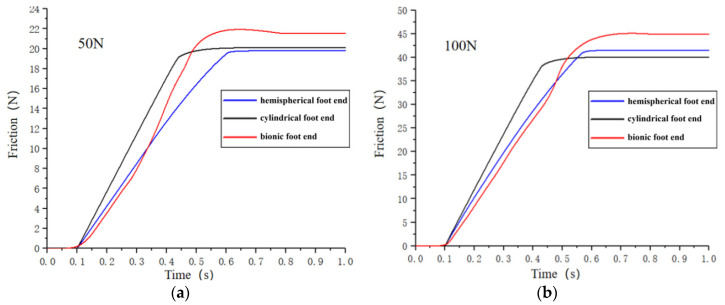
Friction between three types of foot ends and the rock ground at different pressures. (**a**) Pressure of 50 N; (**b**) pressure of 100 N; (**c**) pressure of 150 N; (**d**) pressure of 200 N.

**Figure 11 biomimetics-09-00260-f011:**
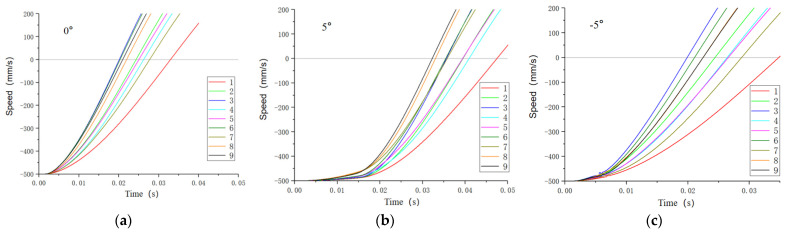
Variation in metacarpal velocities under different touchdown angles. (**a**) Touchdown angle 0°; (**b**) touchdown angle 5°; (**c**) touchdown angle −5°.

**Table 1 biomimetics-09-00260-t001:** Hard rubber material parameters.

Materials	C10 (MPa)	C01 (MPa)	Poisson’s	Densities (1 × 10^−9^ t/mm^3^)	Shore Hardness (HA)
Hard rubber	0.5792	0.1448	0.5	1.112	60

**Table 2 biomimetics-09-00260-t002:** Different ground performance parameters.

Ground Type	Densities (t/mm^3^)	Young’s Modulus E (MPa)	Poisson’s	Friction Angle (°)	Stress Ratio	Expansion Angle (°)
Soil	1.79 × 10^−9^	1.14	0.3	12.21	1	0
Rock	2.72	75,000	0.3	51.8	29.5	0

**Table 3 biomimetics-09-00260-t003:** Coefficient of friction between foot end and different surfaces.

Ground Type	Soil	Rock
coefficient of friction	0.25	0.4

**Table 4 biomimetics-09-00260-t004:** Aluminum alloy material parameters.

Densities (g/cm^3^)	Modulus of Elasticity (GPa)	Poisson’s	Tensile Strength (MPa)	Yield Strength (MPa)
2.7	70	0.3	290	240

**Table 5 biomimetics-09-00260-t005:** Experimental factor and levels.

Factor	z1 (N/mm)	z2 (N/mm)	z3 (N/mm)
Level			
1	50	50	50
2	100	100	100
3	150	150	150

**Table 6 biomimetics-09-00260-t006:** Test results.

Test Number	Factor			0° Impact Time (s)	5° Impact Time (s)	−5° Impact Time (s)
	z1	z2	z3			
1	1 (50)	1 (50)	1 (50)	0.0329	0.0474	0.0348
2	1	2 (100)	2 (100)	0.0243	0.0394	0.0244
3	1	3 (150)	3 (150)	0.0201	0.0356	0.0199
4	2 (100)	1	3	0.0262	0.0409	0.0263
5	2	2	1	0.025	0.0394	0.0265
6	2	3	2	0.0203	0.0354	0.0211
7	3 (150)	1	2	0.0277	0.0358	0.0287
8	3	2	3	0.0219	0.033	0.0225
9	3	3	1	0.021	0.0323	0.0224

## Data Availability

The data that support the findings of this study are available on reasonable request from the corresponding author. The data were not publicly available because of privacy or ethical restrictions.

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
