# Peer review of "Design of Bionic Foot Inspired by the Anti-Slip Cushioning Mechanism of Yak Feet"

_biomimetics, 2024, doi:10.3390/biomimetics9050260_

Round 1
Reviewer 1 Report
Comments and Suggestions for Authors
The paper investigates the anti-slip cushioning mechanism of the Yak’s foot in the design of a bionic foot. Notably, while the introduction and the subsequent literature mentions robotics, this is not referred to in the abstract. The paper refers to bionics at various points, as the biological study, yet in the conclusion biomimetic is used, causing confusion in the author’s use of terminology. The rationale behind selecting the yak as the bioinspired model also lacks a systematic explanation, subsequently resulting in a lack of clarity on how the authors have arrived at the Yak for their bioinspired example.
Overall, the writing could be much improved. This includes:
- Check spacing and punctuation throughout.
- Check that the paper is written in past tense.
- Do also make sure that sentences are not too long, and paragraphs are around 5-6 sentences in length.
I also have specific comments on the science of the paper:
Introduction
- Paragraph 3 is just a list of examples. These examples need to be explored properly to show how the field has expanded, or what has been learned from these.
- Paragraph 4 – state some use-cases where robots need to be on a mountain
- I think the Yak example needs a better justification. Perhaps thinking about a heavy robot is a good first step, but there are still other animals that could be justified here. This area definitely needs work. The work is not justified as it stands.
Materials and Methods
- All dissection figures should be combined into one multipanel figure and there should be more descriptions in the captions. You don’t actually present any dissection findings in the results, so be clear that this is purely methodological inspiration, and how this was used to inform the model etc.
- Subtitle says you did histology, but you did not do any histology, so remove throughout.
- Specifications of the CT scanner and the MRI scanner should be integrated into the text, as well as the respective company details of both the Solidworks and Abaqus software.
- Table 1, outlining hard rubber parameters, could benefit from the inclusion of the shore hardness value, considering its potential impact on the conducted tests.
- Section 2.4 is convoluted and would benefit from streamlining for improved clarity and conciseness. Particularly, the elongated single sentence at line 215 to 226, which should be restructured into more digestible segments, because it struggles to explain the work.
- Phrase on lines 229 and 230 ‘by their compression 229 and deformation’ is repeated, whilst lines 242 to 250 could be rephrased for enhanced comprehension of the described process.
- You have two control shapes, but the bionic foot is quite complex. It is quite a big jump from the simple shapes to the foot, and it is not possible to say what aspect of the bionic foot, specifically, affects the results. This need justifying and explaining in more detail – why you decided to take an approach without systematically varying the bionic parameters.
- Figure 2(b) seems be cropped missing some of the annotation, and the alignment of Figure 4 on the page suggests there is an image missing.
Results
The results show a positive correlation between the three examples of foot ends with time of slip and the maximum static friction in both the soil and rocky scenarios, with graphs to support this. The Regression analysis applied in this study seems suitable, referencing the sole of the foot at various angles. The explanation is clear, supported by equations for justification.
- The results section is nicely presented, with stress cloud diagrams in Figure 11 serving as effective visual aids. However, enlarging these images could display the results better.
- The lengthy sentence structure may hinder the comprehension of the text and should be revisited. At times disjointed, lacking adequate breaks within paragraphs, thereby hindering reader comprehension of the text.
You should include a discussion. The results should be discussed fully in depth and compared to the literature. Limitations and future recommendations should also be included.
Comments on the Quality of English Language
I provided some writing tips above, but I feel uncomfortable saying this is english language quality and don't really agree with having this as a box that I need to fill in.
Author Response
Greetings, esteemed reviewer! We have revised the article in response to your suggestions.Please see the attachment.

Reviewer 2 Report
Comments and Suggestions for Authors
1. The font size in several figures is excessively small.
2. How was friction controlled for different foot shapes considering the relationship between friction, contact area, and force?
3. Figures 9 and 10 should include a view of the lower part of the foot bone structure since it significantly affects friction.
4. The abstract states that the curved shape of the yak sole exhibits higher maximum static friction than the traditional foot, and the conclusion claims that the yak-like foot sole demonstrates superior anti-slip performance compared to the traditional foot. However, no precise comparison with the traditional foot (as described in published papers or in this paper) is presented. While the paper employs cylindrical and spherical simplifications, it could have included a comparison with previous publications on bio-inspired robotic foot designs.
5. The research should compare the proposed bio-inspired method with existing literature on bionic foot design, highlighting its advantages or similarities to existing approaches.
Author Response
Greetings, esteemed reviewer! We have revised the article in response to your suggestionsPlease see the attachment.

Round 2
Reviewer 1 Report
Comments and Suggestions for Authors
Thank you for taking on board my comments. I have just a few things in relation to the new changes that I suggest:
- You have provided more detail in the introduction, however this is very repetitive and you have referred to the studies in an odd way. Always use the same formatting - e.g. Cheng et al. - do not add more names or places of work to this citation. Try to build in a story to identfy where the knowledge gaps are, rather than just saying "they did this".
- Thank you for now incuding a discussion. I would suggest starting with your own study, and then leading on to the more general things. You need to put your findings in the context of the literature - so the paragraph on your results should try to incorporate citations that show other work that is similar or different to your own.
Comments on the Quality of English Language- Although you said that you had checked the spacing and punctuation, there are even more errors than before. After all punctuation there needs to be a space. Please check throughout.
- Please also read through and make sure the language is as clear as it can be. Sometimes it is a little unclear or grammatically incorrect.
Author Response
Thank you for your suggestions, we have revised the article in response to the comments!Please see the attachment

Reviewer 2 Report
Comments and Suggestions for Authors
The authors have modified the paper according to the reviewer's comments.
Author Response
Dear reviewer, thank you for the advice you gave!